# Language Models as Feature Extractors for Accurate Continual Learning

## Abstract

This paper addresses the challenges of *class incremental learning* (CIL) within the broader context of continual learning. In CIL, a system learns a sequence of tasks or classes incrementally. The resulting classifier can categorize test samples into any learned class thus far without relying on task-specific information during testing. CIL presents two significant challenges: *catastrophic forgetting* (CF) and *inter-task class separation* (ICS). ICS occurs because the system lacks data from previous tasks when learning new ones, making it harder to establish decision boundaries between classes, reducing accuracy. This paper proposes a novel method to overcome both CF and ICS. The basic classifier is based on the statistical technique *Mahalanobis distance* (MD), which measures the distance of a data point to a normal distribution. In the proposed approach, each class is represented by a normal distribution with the mean and covariance derived from the features of its training data, which are extracted from a language model (LM). To reduce storage, all classes share a common covariance matrix. Two additional techniques are also proposed to enhance the accuracy: (1) using a kernel function to expand the feature space, and (2) incorporating an ensemble mechanism. Our experiments show that the proposed method achieves accuracy comparable to the upper bound accuracy of joint fine-tuning, which, to our knowledge, has not been achieved before.

## 1 Introduction

The objective of continual learning (CL) is to enable an AI agent to incrementally acquire knowledge by learning a sequence of tasks over time (Chen & Liu, 2018; Wang et al., 2023a). One of the major challenges in CL is *catastrophic forgetting* (CF), where updating the model's parameters to learn new tasks can lead to a decline in performance on previously learned tasks (McCloskey & Cohen, 1989). In this work, we focus on the *class incremental learning* (CIL) setting of continual learning (CL) (Van de Ven & Tolias, 2019).

In CIL, each task introduces a unique set of classes, and the goal is to train a single unified model capable of recognizing all the classes encountered across tasks. Since learning has to be done incrementally, learning a task has no or limited access to the data of previous tasks. A defining characteristic of CIL is that no task identification information is provided during testing, meaning the model does not know which task a test sample belongs to.[1] An additional challenge to CIL is *inter-task class separation* (ICS) (Kim et al., 2022), which refers to the problem that without access to previous task data when learning a new task, it is difficult to establish decision boundaries between new and previously learned classes. While many CIL methods have been proposed (De Lange et al., 2021; Zhou et al., 2024), they still suffer from significant performance degradation as more tasks are learned, largely due to the combined challenges of CF and ICS.

This paper proposes a novel technique, called *Kernel Mahalanobis Distance* (KMD), to address both challenges and can achieve an accuracy level comparable to that obtained by joint training of all classes/tasks together, which is regarded as the accuracy upper bound of CIL, with no replay data.

---

[1]Two other common CL settings are *task incremental learning*, where task information is provided during testing, and *domain-incremental learning*, which involves learning tasks from different domains but with the same set of classes (Van de Ven & Tolias, 2019).

KMD leverages a *frozen* language model (LM) as a fixed feature extractor, meaning that during learning, the LM's parameters remain unchanged, and no additional structures like learnable prompts (Wang et al., 2022b; Razdaibiedina et al., 2023) or adapters (Houlsby et al., 2019) are added to tailor it for each new task. KMD operates under the assumption that each class follows a Gaussian distribution characterized by a mean and covariance. It further assumes that all classes share the same covariance but differ in their means. Throughout the CIL process, KMD simply calculates and updates the shared covariance and class-specific means based on the features extracted by the LM for each training sample. For classification, it employs Mahalanobis Distance (MD)–a function that measures the distance between a data point and a Gaussian distribution–to compare the test sample to the distribution of each class and select the nearest match. Since the approach involves no training or updates to the LM's parameters, it avoids CF caused by parameter updates.

Two additional enhancements are proposed to further improve the results. First, employing the Radial Basis Function (RBF) kernel (Scholkopf & Smola, 2018) to improve the features extracted from the LM. The RBF kernel computes the dot product in an implicit higher-dimensional space, giving higher similarity to closer points and lower similarity to those further apart. This creates a localized effect, where the influence of each data point diminishes as you move further away from it, which is desirable for our distance-based classification approach. However, the RBF kernel requires storing the pairwise similarities for all data points, which is infeasible. To address this, We approximate the kernel using *Random Fourier Features* (RFF) (Rahimi & Recht, 2007), making it feasible for CIL. When a new task arrives, KMD uses the *kernalized* features to compute the class *mean* of each class and updates the *shared covariance matrix* for all classes learned thus far. Second, instead of relying on a single kernel transformation, we create a few parallel models and perform an ensemble for the final classification, which gives the final system KMD-Ensemble. In summary, this paper makes three key contributions:

(1) It presents a novel CIL method called KMD, which leverages the rich features extracted from an LM, further enhancing them through a kernel technique and an ensemble strategy. To our knowledge, this approach has not been previously explored.

(2) KMD addresses both CF and ICS challenges by (i) using a shared covariance matrix and class feature means to define Gaussian distributions, effectively addressing ICS by creating clear class boundaries, and (ii) gathering only statistical information during CIL without updating the LM or training new networks, thus avoiding CF at the LM level. Although incrementally updating the covariance matrix could introduce CF, our results show even a slight positive knowledge transfer.

(3) Our experiments show that KMD-Ensemble significantly outperforms existing baselines, achieving accuracy comparable to joint Fine-tuning of the LM on all tasks, the upper bound for CIL. We further validate this using various LMs. This is notable, as existing CIL methods have consistently below this upper bound–often by a considerable margin–posing a significant barrier to their practical use in real-world applications.

## 2 RELATED WORK

There is a large body of literature on continual learning (CL). Most of it focuses on mitigating catastrophic forgetting (CF). Existing methods generally fall into several categories: *Regularization-based methods* use regularizers to penalize changes to important parameters of previous tasks, (Kirkpatrick et al., 2017; Zenke et al., 2017; Li et al., 2022; Liu et al., 2019a). *Replay-based methods* store some samples from previous tasks and learn a new task using both the new and stored data to maintain performance across tasks (Liu et al., 2021a; Scialom et al., 2022; Qin et al., 2022; Huang et al., 2021). Some methods learn data generators instead of storing actual data to generate samples similar to those from previous tasks (Shin et al., 2017; He & Jaeger, 2018). *Architectural-based methods* include many approaches. Some expand the network as new tasks are introduced (Wang et al., 2022a; Yan et al., 2021; Qin et al., 2023). Some use *parameter isolation*, where sub-networks are trained for each task via mechanisms like masking or orthogonal projection (Serra et al., 2018; Gururangan et al., 2021; Zhu et al., 2022; Geng et al., 2021; Lin et al., 2022; Wortsman et al., 2020; Liu et al., 2023). Yet, some class incremental learning (CIL) methods employ a task predictor to identify the appropriate model for the predicted task classifier (Rajasegaran et al., 2020; Abati et al., 2020; Wang et al., 2023b; 2024a). They may utilize strategies like separate networks, entropy, or out-of-distribution (OOD) detection to determine the task and to deal with inter-task class separa-

tion (ICS) problem. Most CL methods operate in a batch setting, where all task data is available at once. There are also online CL methods that process data incrementally in a stream (Mai et al., 2022). While we do not focus on streaming, our approach shares similarities with (Hayes & Kanan, 2020), which uses *streaming linear discriminant analysis* with class means and a covariance matrix. However, our work differs by employing Mahalanobis distance. Additionally, we enhance feature representations with a kernel and utilize an ensemble technique.

In natural language processing (NLP), continual learning has been applied to a wide range of problems, e.g., text classification (Sun et al., 2020; Chuang et al., 2020), sentiment analysis (Ke et al., 2021), topic modeling (Gupta et al., 2020), slot filling (Shen et al., 2019), question answering (Greco et al., 2019), language acquisition (Li et al., 2019; Liang et al., 2024; Zhao et al., 2024), and the pre-training of language models (Qin et al., 2022; Ke et al., 2021). Employing pre-trained models is a standard approach in many NLP-related CL scenarios as leveraging their capabilities can improve performance (Shao et al., 2023; Wang et al., 2024b). For further insights and overview, please refer to the surveys (Ke & Liu, 2022; Wang et al., 2023a).

The rise of large foundation models has led to a growing interest in integrating CL with pre-trained models (Yang et al., 2024). While prior work incorporates pre-trained models, they do so within the framework of the three main CL strategies discussed above, which still suffer from CF. In contrast, we explore the full potential of the pre-trained large language models as fixed feature-extractors for CIL, i.e., leveraging only their latent features for downstream tasks. However, our experiments show that using latent features directly is sub-optimal. To address this, we propose KMD, which enhances features with kernel functions, improving class separability in the kernelized space. KMD is distinct in being replay-free, without relying on regularizers or architectural changes.

# 3 BACKGROUND

This section presents the main background information for the proposed method.

**Class Incremental Learning (CIL):** In CIL, a model is trained on a sequence of tasks $\{\mathcal{T}_1, \mathcal{T}_2, \ldots, \mathcal{T}_T\}$, where each task $\mathcal{T}_t$ introduces a disjoint set of classes with its associated training data $\mathcal{D}_t = \{(x_t^{(i)}, y_t^{(i)})\}_{i=1}^{N_t}$. The learning process is incremental, meaning that the data from previous tasks $\mathcal{T}_1, \ldots, \mathcal{T}_{t-1}$ is not accessible while learning the current task $\mathcal{T}_t$. The model must learn new classes without forgetting previously learned ones, despite the absence of earlier training data. The goal is to learn a unified model $F : \mathcal{X} \to \mathcal{Y}$ capable of classifying samples from any of the classes encountered across the $T$ tasks. During inference, the task identity is unknown, and the model must predict the correct class label from all the classes encountered so far.

## 3.1 CLASS-PROTOTYPES FOR CONTINUAL LEARNING

Fine-tuning an LM for CIL often leads to *catastrophic forgetting* (CF). Instead, leveraging the extracted features from a powerful LM to incrementally accumulate *class-prototypes* (CPs) while keeping the LM frozen can result in more accurate classification, as we will demonstrate in Section 5.5.1. A simple yet effective method is the Nearest Class Mean (NCM) classifier, where the prototype for each class is the *mean* of the feature vectors extracted from the LM for all training samples of that class. For simplicity, from this point on, we use $\mathbf{x}$ to denote the feature vector extracted from the LM for an input $x$. The class prototype $\mu_m$ for a class $m$ is computed as:

$$\mu_m = \frac{1}{n_m} \sum_{i=1}^{n_m} \mathbf{x}_i \tag{1}$$

where $n_m$ is the number of samples for class $m$. This mean vector can be computed incrementally for each class and does not cause CF as it doesn't involve any training. During inference, a test sample is classified by finding the class mean with the highest *cosine similarity*.

$$\hat{y} = \arg\max_m \frac{\mathbf{x}_{\text{test}}^\top \mu_m}{\|\mathbf{x}_{\text{test}}\| \|\mu_m\|} \tag{2}$$

This straightforward method surprisingly outperforms more complex prompt-based, generation-based, or Fine-tuning-based CIL baselines (see Section 5.5.1), which are susceptible to CF. This suggests that LMs provide robust, generalizable representations suitable for downstream tasks.

To enhance NCM, higher-order statistics can be incorporated. We can represent the data by a multivariate Gaussian distribution $\mathcal{N}(\mu_m, \boldsymbol{\Sigma}_m)$, where each class has its own mean $\mu_m$ and covariance $\boldsymbol{\Sigma}_m$. The *Mahalanobis distance* (MD) (De Maesschalck et al., 2000) is then used for classification, where a test sample is assigned to the class whose distribution is closest in terms of MD.

However, storing a separate covariance matrix $\boldsymbol{\Sigma}_m$ for each class becomes impractical in a continual learning setting, as the number of parameters grows significantly with the introduction of new classes. To address this, we assume that all classes share the same covariance matrix $\boldsymbol{\Sigma}$, which allows us to keep the model tractable and suitable for continual learning. The shared covariance matrix is computed as follows:

$$\boldsymbol{\Sigma} = \frac{1}{N} \sum_{m=1}^{M} \sum_{i=1}^{n_m} (\mathbf{x}_{m,i} - \mu_m)(\mathbf{x}_{m,i} - \mu_m)^\top \tag{3}$$

where $M$ is the number of classes seen so far, and $N$ is the total number of samples, i.e., $N = \sum_{m=1}^{M} n_m$. This shared covariance matrix can be updated incrementally during the CIL process as new tasks arrive by first computing the mean for each class before updating $\boldsymbol{\Sigma}$. Under the shared covariance assumption, the Mahalanobis distance for classification can be written as:

$$\text{MD}(\mathbf{x}_{\text{test}}, \mu_m, \Sigma) = \sqrt{(\mathbf{x}_{\text{test}} - \mu_m)^\top \boldsymbol{\Sigma}^{-1} (\mathbf{x}_{\text{test}} - \mu_m)} \tag{4}$$

This method addresses some limitations of the NCM method by taking into account the covariance of the data, while avoiding the excessive parameter overhead of storing individual covariance matrices for each class.

## 4 PROPOSED METHOD: KMD

We propose enhancing MD with a kernel function, resulting in the KMD method. The core idea is to improve the separability of feature representations obtained from the pre-trained LM using a kernel function. This allows us to maintain a robust representation of the data as new classes/tasks are added, without suffering from CF.

### 4.1 KERNEL FUNCTIONS

While MD works well for separable Guassian distributions, it may struggle when class boundaries are not well-separated in the original feature space. A powerful approach to overcome this limitation is through the use of kernel functions, which implicitly map the input data (features from the LM in our case) into a higher-dimensional space where the data becomes more separable. In this high-dimensional space, MD can provide better class separation, even when the original feature space lacks clear boundaries.

Mathematically, if we have an input space $\mathcal{X}$ and a mapping $\varphi : \mathcal{X} \to \mathcal{V}$, where $\mathcal{V}$ is a potentially infinite-dimensional feature space, a kernel function $K(\mathbf{x}_i, \mathbf{x}_j)$ computes the inner product in this space without explicitly performing the transformation:

$$K(\mathbf{x}_i, \mathbf{x}_j) = \langle \varphi(\mathbf{x}_i), \varphi(\mathbf{x}_j) \rangle_{\mathcal{V}} \tag{5}$$

One of the most commonly used kernels is the Radial Basis Function (RBF) kernel (Scholkopf & Smola, 2018), which is defined as:

$$K(\mathbf{x}_i, \mathbf{x}_j) = \exp\left(-\frac{\|\mathbf{x}_i - \mathbf{x}_j\|^2}{2\sigma^2}\right) \tag{6}$$

The RBF kernel corresponds to an inner product in an infinite-dimensional space, making it highly effective for capturing complex patterns in data.[2] However, directly computing the kernel matrix $\mathbf{K}$ for all pairs of instances in a dataset of size $N$ leads to a large matrix of size $N \times N$, which is computationally prohibitive. In the continual learning setting, this approach is infeasible as it requires access to data from all previous tasks, which is not available.

---

[2] We also experimented with several other kernel functions and found the RBF kernel to be better suited for our CIL setup.

## 4.2 Approximating the Kernel with Random Fourier Features

To address the challenges associated with the kernel method in the CIL setting, we approximate the kernel function using Random Fourier Features (RFF) (Rahimi & Recht, 2007). This method is grounded in Bochner's theorem (Rudin, 2017), which states that any continuous, shift-invariant kernel can be represented as the Fourier transform of a non-negative measure:

$$K(\mathbf{x}_i, \mathbf{x}_j) = \int p(\omega) e^{i\omega^\top(\mathbf{x}_i - \mathbf{x}_j)} d\omega = \mathbb{E}_\omega \left[ e^{i\omega^\top(\mathbf{x}_i - \mathbf{x}_j)} \right] \tag{7}$$

Here, $\omega$ is the frequency in the Fourier domain, and $p(\omega)$ is the probability density function associated with $\omega$. Given that both the kernel $K(\mathbf{x}_i, \mathbf{x}_j)$ and the distribution $p(\omega)$ are real, the integral can be simplified. The complex exponential $e^{i\omega^\top(\mathbf{x}_i - \mathbf{x}_j)}$ can be expressed in terms of its real part using Euler's formula. Therefore, we can obtain a real-valued mapping that satisfies the condition $\mathbb{E}[z_\omega(\mathbf{x_i}) z_\omega(\mathbf{x_j})] = K(\mathbf{x_i}, \mathbf{x_j})$ by setting:

$$z_\omega(\mathbf{x}) = \sqrt{2} \cos(\omega^\top \mathbf{x} + \beta) \tag{8}$$

where $\omega \sim p(\omega)$, $\beta \sim \text{Uniform}(0, 2\pi)$. For the RBF kernel, the Fourier transform $p(\omega)$ is a Gaussian distribution (Rahimi & Recht, 2007). We now have a simple and efficient algorithm to estimate the kernel function by pooling $D$ independent pairs $\omega$, $\beta$ from these distributions and estimating the expectation. Therefore, we can define the random feature map as:

$$\mathbf{z}(\mathbf{x}) = \sqrt{\frac{2}{D}} \left[ \cos(\omega_1^\top \mathbf{x} + \beta_1), \ldots, \cos(\omega_D^\top \mathbf{x} + \beta_D) \right] \tag{9}$$

where $\omega$ is drawn from $\mathcal{N}(0, \sigma^{-2}\mathbf{I})$ and $\beta$ from $\text{Uniform}(0, 2\pi)$. As the number of pooled pairs $D$ increases, the approximation of the kernel function improves because more Monte Carlo samples are used to estimate the expectation. The dot product of these random features approximates the original kernel function:

$$\mathbf{z}(\mathbf{x_i})^\top \mathbf{z}(\mathbf{x_j}) \approx K(\mathbf{x_i}, \mathbf{x_j}) \tag{10}$$

Thus, $\mathbf{z}$ represents an approximation of $\varphi$. We can now convert the input $\mathbf{x}$ into random features $\mathbf{z}(\mathbf{x})$ and apply MD. This approximation enables us to avoid directly computing the kernel matrix, making it feasible to apply in continual learning settings while preserving the benefits of the kernel transformation.

## 4.3 Classification with KMD and Ensembles

**Training:** The training process for KMD is outlined in Algorithm 1. We first apply RFF to the original feature vector $\mathbf{x} \in \mathbb{R}^d$, transforming it into $\mathbf{z} \in \mathbb{R}^D$. With each new class, the mean $\mu_m$ is calculated, and the shared covariance matrix $\Sigma$ is updated incrementally.

**Inference:** Given a test sample, we apply RFF to obtain its transformed representation $\mathbf{z}_{\text{test}}$. The MD is then computed between $\mathbf{z}_{\text{test}}$ and the distribution of each class $m$ (i.e., $(\mu_m, \Sigma)$). The predicted class is the one with the smallest distance:

$$\hat{y} = \arg\min_m \text{MD}(\mathbf{z}_{\text{test}}, \mu_m, \Sigma) \tag{11}$$

**Ensemble Method:** To enhance performance, we introduce an ensemble mechanism, leading to our final system **KMD-Ensemble**. Multiple KMD models are trained using different RFF transformations, each initialized with distinct frequency matrices and phase vectors (see Algorithm 1). During inference, the MDs are calculated for each model, then negated and transformed into probabilities via the **softmax** function:

$$P(y = m \mid \mathbf{z}_{\text{test}}) = \frac{\exp(-\text{MD}(\mathbf{z}_{\text{test}}, \mu_m, \Sigma))}{\sum_{c=1}^{M} \exp(-\text{MD}(\mathbf{z}_{\text{test}}, \mu_c, \Sigma))} \tag{12}$$

The final prediction is made by averaging the probabilities across all $E$ models, with the class having the highest average probability selected: [3]

$$\hat{y} = \arg\max_m \frac{1}{E} \sum_{e=1}^{E} P_e(y = m \mid \mathbf{z}_{\text{test}}) \tag{13}$$

---

[3]We also experimented with other ways of combining predictions from multiple models, such as hard voting, but found this approach to be the most effective, as it resolves tie votes.

---

**Algorithm 1** KMD Training

---

1: **Initialize**
2: $\omega \sim \mathcal{N}(0, \sigma^{-2}\mathbf{I}) \in \mathbb{R}^{d \times D}$                         {RFF frequency matrix}
3: $\beta \sim U(0, 2\pi) \in \mathbb{R}^{D}$                                {RFF phase vector}
4: $\boldsymbol{\Sigma} = \mathbf{0} \in \mathbb{R}^{D \times D}$                           {shared covariance matrix}
5: $N_{\text{total}} = 0$                                {total number of samples}

6: **Function** RFF($X$):
7:     **return** $\sqrt{\frac{2}{D}} \cos(X\omega + \beta)$                      {RFF applied to batch}

8: **Function** Update($X, m$):
9:     **Input:** $X \in \mathbb{R}^{n_m \times d}$ - batch feature vectors for all training samples of class $m$
10:     $N_{\text{prev}} \leftarrow N_{\text{total}}$
11:     $N_{\text{total}} \leftarrow N_{\text{total}} + n_m$
12:     $Z \leftarrow \text{RFF}(X)$
13:     Compute class mean: $\mu_m \leftarrow \frac{1}{n_m} \sum_{i=1}^{n_m} Z_i$
14:     Update covariance matrix:
15:         $\boldsymbol{\Sigma} \leftarrow \frac{N_{\text{prev}}}{N_{\text{total}}} \boldsymbol{\Sigma} + \frac{1}{N_{\text{total}}} \sum_{i=1}^{n_m} (Z_i - \mu_m)(Z_i - \mu_m)^{\top}$

---

## 5 EXPERIMENTAL EVALUATION

To evaluate our proposed method, KMD, we conduct experiments across multiple text classification datasets and compare KMD against different types of baselines. The code of KMD has been submitted in *Supplementary Materials*.

### 5.1 DATASETS

Four text classification datasets are used in our experiments: **1. CLINC**: This dataset has 150 classes, which are dialogue intents, from many different application domains (Larson et al., 2019). We used the train/test split of 10,000/750 samples, and the classes were randomly divided into 10 disjoint tasks. **2. Banking**: This dataset has 77 classes of dialogue intents in the banking domain (Casanueva et al., 2020). We employed a 10,000/1,000 train/test split and divided the classes into 7 disjoint tasks. **3. DBpedia**: A text classification dataset of Wikipedia articles with 70 classes (Liu et al., 2021b). We used a train/test split of 10,000/1,000 samples and divided the classes into 7 disjoint tasks. **4. HWU**: Another dialogue intent classification dataset with 20 domains and 64 classes (Auer et al., 2007). We used a train/test split of 9,000/1,000 samples and partitioned the classes into 8 disjoint tasks.

We adhere to the standard CIL protocol, where the classes are partitioned into disjoint tasks. The classes within each dataset are randomly shuffled and assigned to these tasks, ensuring that each task introduces new classes not seen in previous tasks. To account for the variability in performance due to different task splits, we perform multiple runs with different random shuffles and report the average results.

### 5.2 BASELINES

We compare KMD against a range of baselines, categorized into existing CIL methods, class-prototype based methods, and joint training. These diverse baselines allow us to thoroughly evaluate the performance of KMD.

– *Existing CIL Baselines*: These systems use various existing popular approaches. **1. Vanilla**: Sequentially fine-tunes the model on each task with no mechanism to mitigate CF. **2. EWC (Elastic Weight Consolidation)**: A popular regularization-based method that adds a penalty to preserve important parameters from previous tasks, balancing new learning with retention (Kirkpatrick et al., 2017). **3. KD (Knowledge Distillation)**: Uses knowledge distillation to help the model retain information from old tasks by learning from softened output probabilities of previous versions of itself (Hinton et al., 2015). **4. L2P (Learn to Prompt)**: Freezes the LM and learns trainable prompts to guide inference, adapting to new tasks without altering the LM (Wang et al., 2022b). **5. LAMOL**

**(Language Modeling for Lifelong Language Learning)**: Employs pseudo-replay by generating pseudo-examples of previous tasks to mix with new task data, maintaining past performance while learning new tasks (Sun et al., 2019). **6. VAG (Vocabulary-Aware Label Generation)**: Leverages vocabulary sparsity to selectively activate relevant outputs for each task, mitigating forgetting. Instead of traditional classification, VAG focuses on generating labels (Shao et al., 2023).

– *Class-Prototype Based Baselines:* **7. NCM (Nearest Class Mean)**: Maintains a mean feature vector for each class, updated incrementally. Classification is based on the nearest class mean to the test sample's feature vector. **8. MD (Mahalanobis Distance)**: Uses the original feature space **without** our RBF kernel extension.

– *Upper Bound Baseline:* **9. Joint Fine-tuning**: Fine-tuning the full LM by adding a classifier head on top of the latent features and training on all classes simultaneously as a single task. The result from this is considered the **upper-bound performance** of CIL.

## 5.3 IMPLEMENTATION DETAILS

For all experiments–except the ablation on different LMs–we use the BART-base model (Lewis et al., 2019), which features a 6-layer encoder-decoder architecture with a 768-dimensional hidden state. We chose this model because many of our baselines employ a generative objective or require generating pseudo-replay data during training, making the decoder component essential. Additionally, the state-of-the-art baseline VAG (Shao et al., 2023) also utilizes BART-base.

To study the generalization of our method to other LMs, we also evaluate it using the following models: **1. paraphrase-MiniLM-L3** (Reimers & Gurevych, 2019) (3 layers, 384 dimensions), **2. BERT-base** (Devlin, 2018) (12 layers, 768 dimensions), **3. RoBERTa-large** (Liu et al., 2019b) (24 layers, 1024 dimensions), **4. T5-3b** (Raffel et al., 2020) (24 layers, 1024 dimensions), and **5. Mistral-7b** Jiang et al. (2023) (32 layers, 4096 dimensions).

LAMOL and VAG were executed using their official codes and configurations. For the remaining existing baselines, we used implementations from (Shao et al., 2023) repository. The class-prototype based baselines were implemented using our own code, adhering to the same update rules applied in KMD to ensure consistency in comparison.

The **Joint Fine-tuning** model, regarded as the upper bound, is trained for 50 epochs with a batch size of 128, using the Adam optimizer with a learning rate of 1e-3 for the classifier head and 1e-4 for the LM parameters. Additionally, we experimented with various configurations, including different learning rates, batch sizes, and epoch numbers, to ensure the models were *thoroughly trained and optimized*. We also compared our Joint Fine-tuning results with those reported in (Shao et al., 2023) and found that our configuration actually achieves better performance.

For KMD-Ensemble, we always use an ensemble of 5 models, but we also provide an ablation study on the impact of the number of models in the ensemble. KMD itself has two hyperparameters: the transformation dimension $D$ and the RFF $\sigma$. Given the CIL setup, where tasks are learned incrementally, the system does not see all tasks at the same time, and validation sets are not typically available. Therefore, it is hard to optimize the parameters for all tasks. Through empirical testing, we found that setting $D$ to 5000 offers a balanced trade-off between memory usage and performance. The $\sigma$ parameter is also empirically determined within range $[10^{-2}, 10^{-6}]$ for each LM and remains fixed across different datasets. We will show the results of different parameter settings later. Our implementation is built using PyTorch, with all LMs sourced from the Hugging Face Transformers library. All experiments are conducted on a NVIDIA A100 GPU with 80GB of VRAM.

## 5.4 EVALUATION METRIC

We measure classification accuracy after all tasks have been processed, referred to as **Last or Final Accuracy**. Additionally, we use **Forgetting Rate** to quantify how much the model forgets previously learned tasks as it learns new ones. For each task $i$, let's define $A_i^t$ as the accuracy on the data of task $i$ after learning task $t$, where the classification is restricted to classes of task $i$. By this definition, $A_i^i$ represents the model's accuracy on task $i$ immediately after learning it, serving as the initial performance benchmark for the task. The *Forgetting Rate* after learning task $t$, denoted as $F_t$, is calculated as the average loss in accuracy across all tasks up to $t$: $F_t = \frac{1}{t} \sum_{i=1}^{t} \left( A_i^i - A_i^t \right)$. A

higher value of $F_t$ indicates greater forgetting, while a lower value suggests that the model retains information from previously learned tasks more effectively. We also discuss the efficiency and the memory requirement of the proposed method.

| Method | CLINC (10-T) | Banking (7-T) | DBpedia (7-T) | HWU (8-T) |
|---|---|---|---|---|
| Joint Fine-tuning | 95.33 $\pm_{0.04}$ | 91.36 $\pm_{0.32}$ | 94.83 $\pm_{0.16}$ | 88.60 $\pm_{0.29}$ |
| Vanilla | 42.06 $\pm_{1.53}$ | 31.80 $\pm_{1.20}$ | 43.45 $\pm_{2.54}$ | 30.95 $\pm_{3.37}$ |
| EWC | 45.73 $\pm_{0.46}$ | 38.40 $\pm_{2.70}$ | 44.99 $\pm_{2.90}$ | 34.01 $\pm_{3.46}$ |
| KD | 36.33 $\pm_{0.86}$ | 27.40 $\pm_{1.59}$ | 42.10 $\pm_{2.40}$ | 25.46 $\pm_{2.13}$ |
| L2P | 30.66 $\pm_{2.46}$ | 31.45 $\pm_{0.55}$ | 23.52 $\pm_{1.54}$ | 24.04 $\pm_{0.88}$ |
| LAMOL | 58.42 $\pm_{0.84}$ | 42.60 $\pm_{1.36}$ | 48.61 $\pm_{1.82}$ | 44.85 $\pm_{1.57}$ |
| VAG | 76.42 $\pm_{0.90}$ | 59.34 $\pm_{1.28}$ | 65.40 $\pm_{1.52}$ | 56.88 $\pm_{1.22}$ |
| NCM | 83.60 $\pm_{0.00}$ | 71.10 $\pm_{0.00}$ | 75.70 $\pm_{0.00}$ | 73.30 $\pm_{0.00}$ |
| MD | 93.71 $\pm_{0.00}$ | 89.09 $\pm_{0.00}$ | 93.42 $\pm_{0.00}$ | 86.41 $\pm_{0.00}$ |
| KMD | 95.90 $\pm_{0.68}$ | 92.23 $\pm_{0.32}$ | 94.13 $\pm_{0.32}$ | 87.27 $\pm_{1.39}$ |
| **KMD-Ensemble** | **96.62** $\pm_{\mathbf{0.08}}$ | **93.03** $\pm_{\mathbf{0.06}}$ | **94.53** $\pm_{\mathbf{0.12}}$ | **89.78** $\pm_{\mathbf{0.09}}$ |

Table 1: Final accuracy (%) of different methods on various datasets. All results are with a BART-base backbone, and no replay buffer was used for any method. The number of tasks is indicated in parentheses next to each dataset (#-T). Note that the number of tasks does not affect KMD, NCM, or MD, as these methods add a class-prototype at a time. Joint Fine-tuning is considered the upper bound for CIL performance since it learns all classes together as a single task.

## 5.5 RESULTS AND ANALYSIS

We now present and analyze the performance of KMD in comparison with baselines, examining its accuracy, memory usage, and efficiency. We also study how KMD performs across different LMs and its two hyperparameters.

### 5.5.1 COMPARISON WITH BASELINES

Table 1 presents the performance of KMD against various baselines. The existing CIL baselines include EWD, KD, L2P, LAMOL and VAG. Despite specialized mechanisms for mitigating CF, these methods still exhibit significant forgetting, with even the best-performing method, VAG, falling far short of the accuracy achieved by the simple NCM method.

NCM, while effective, significantly underperforms MD and KMD, indicating that merely accumulating a mean feature vector for each class is insufficient to fully leverage the information in the LM's feature representations. KMD improves upon MD by leveraging the kernel method, leading to better performance. The addition of the ensemble approach further enhances accuracy, with KMD-Ensemble outperforming all other methods.

**Joint Fine-tuning Upper Bound.** KMD-Ensemble consistently matches the accuracy of the Joint Fine-tuning upper bound, even surpassing it on 3 out of the 4 datasets, and achieving nearly identical results on the fourth (DBpedia). Notably, even KMD alone performs on par with Joint Fine-tuning. This shows that the features of LMs are well-suited for highly accurate continual learning, and the key lies in how to utilize these features appropriately, which is achieved by the proposed method KMD and KMD-Ensemble for CIL.

### 5.5.2 GENERALIZABILITY ACROSS DIFFERENT LMS

We evaluate KMD-Ensemble's performance across different LMs of varying sizes. The results, shown in Table 2, indicate that KMD-Ensemble consistently achieves performance comparable to or better than Joint Fine-tuning across all datasets, regardless of the LM used. This highlights the robustness of KMD-Ensemble for CIL.

| Method | CLINC | Banking | DBpedia | HWU |
|--------|-------|---------|---------|-----|
| **paraphrase-MiniLM (3 layers, 384 dimensions)** | | | | |
| KMD-Ensemble | $94.53 \pm 0.00$ | $91.73 \pm 0.09$ | $86.83 \pm 0.17$ | $87.95 \pm 0.23$ |
| Joint Fine-tuning | $93.20 \pm 0.16$ | $90.90 \pm 0.08$ | $87.43 \pm 0.16$ | $87.13 \pm 0.12$ |
| **BERT-base (12 layers, 768 dimensions)** | | | | |
| KMD-Ensemble | $94.98 \pm 0.31$ | $91.00 \pm 0.24$ | $95.40 \pm 0.08$ | $88.32 \pm 0.31$ |
| Joint Fine-tuning | $94.56 \pm 0.04$ | $88.96 \pm 0.16$ | $95.03 \pm 0.09$ | $87.26 \pm 0.28$ |
| **RoBERTa-large (24 layers, 1024 dimensions)** | | | | |
| KMD-Ensemble | $96.31 \pm 0.06$ | $92.93 \pm 0.05$ | $94.60 \pm 0.08$ | $89.25 \pm 0.04$ |
| Joint Fine-tuning | $95.96 \pm 0.30$ | $91.16 \pm 0.04$ | $94.99 \pm 0.21$ | $88.40 \pm 0.29$ |
| **T5-3b (24 layers, 1024 dimensions)** | | | | |
| KMD-Ensemble | $96.04 \pm 0.17$ | $93.77 \pm 0.05$ | $95.33 \pm 0.09$ | $89.31 \pm 0.27$ |
| Joint Fine-tuning | $96.86 \pm 0.06$ | $92.30 \pm 0.10$ | $94.60 \pm 0.03$ | $90.30 \pm 0.10$ |
| **Mistral-7b (32 layers, 4096 dimensions)** | | | | |
| KMD-Ensemble | $97.13 \pm 0.11$ | $92.53 \pm 0.12$ | $96.00 \pm 0.08$ | $90.02 \pm 0.09$ |
| Joint Fine-tuning | $97.60 \pm 0.11$ | $92.50 \pm 0.14$ | $95.70 \pm 0.07$ | $90.43 \pm 0.11$ |

Table 2: Comparison of final accuracy (%) between KMD-Ensemble and Joint Fine-tuning across different LMs. Joint Fine-tuning is considered the upper bound for CIL performance since it learns all classes together as a single task.

| Dataset | $F_T$ (%) |
|---------|-----------|
| **CLINC** (10-T) | -0.133 |
| **Banking** (7-T) | -0.105 |
| **DBpedia** (7-T) | -0.085 |
| **HWU** (8-T) | -0.476 |

Table 3: Forgetting rate (%) of KMD on each dataset after learning all tasks. T represents the total number of tasks in each dataset.

### 5.5.3 Analysis of Forgetting Rate

Our method has no forgetting at the LM level since we do not fine-tune it or add any additional structure to it for adaptation. However, the incremental updates to the shared covariance can introduce some forgetting. Therefore, we measured the forgetting rate after all tasks were learned and found it to be negative across all datasets, as shown in Table 3. This indicates slight positive knowledge transfer, meaning the accuracy on earlier tasks improved after learning new ones.

### 5.5.4 Memory Usage Comparison

We compare the methods in terms of memory usage. The Joint Fine-tuning only adds a classifier head on top of the LM features, introducing approximately 0.1M additional parameters for typical values of $M = 150$ classes and $d = 768$ hidden dimensions. Existing method baselines, particularly those requiring the model to operate in the generation mode, significantly increase the memory usage. For instance, an LM head required for text generation adds approximately 38.5M parameters for a vocabulary of 50,265 tokens, although this number does not increase with the number of classes.

Class Prototype (CP) methods are more memory-efficient as they only require storing the class prototypes. NCM requires $M \times d$ parameters for the mean vectors, similar to the classifier head of the Joint model. MD adds an $d \times d$ covariance matrix, increasing the parameter count by approximately 0.6M. KMD introduces $D \times (d + 1)$ fixed non-trainable parameters for the RFF transformation. With $D$ set to 5000, this adds around 3.8M parameters. KMD also scales the parameters required for CPs by a factor of $D/d$, leading to an additional 0.75M parameters. The covariance matrix for KMD is $D \times D$, resulting in an additional 25M parameters. In total, KMD's memory footprint is approximately 29.5M parameters. This memory requirement is still significantly lower than the LM head needed for text generation alone. Our KMD-Ensemble utilizes 5 models, resulting in a 5x increase in memory usage, which remains within a reasonable limit. For reference, the BART-base model used in our experiments has 139.5M parameters. We highlight that a large portion of KMD's parameters are associated with the fixed RFF transformation and the shared covariance matrix, which do not increase as more classes are added in the CIL process.

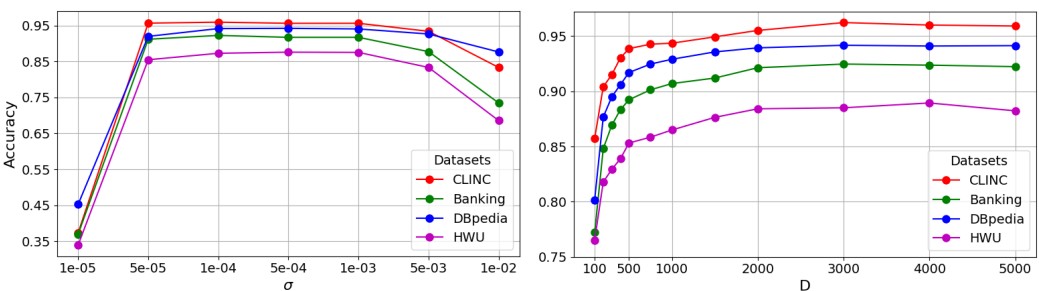

Figure 1: Impact of hyperparameters on KMD performance. (Left) Effect of the RFF parameter $\sigma$ with a fixed transform dimension $D = 5000$. (Right) Effect of varying the transform dimension $D$ with $\sigma = 10^{-4}$.

### 5.5.5 EFFICIENCY AND RUNTIME ANALYSIS

Our method is highly efficient, as it bypasses the need to update LM parameters or compute gradients during training. Instead, it simply computes class means and the covariance matrix. On the CLINC dataset with a BART-base LM, KMD and KMD-Ensemble train in approximately 10 and 30 seconds, respectively, on our GPU setup–comparable to the time required to extract latent features from the LM. In contrast, Joint Fine-tuning takes about 4 minutes to train. Existing CL method baselines take much longer to train, as they involve updating the model incrementally on each task. They often require additional computations too, e.g., computing the output of their previous versions (KD) or generating pseudo-replay data (LAMOL and VAG), leading to training times ranging from 11 to 23 minutes.

### 5.5.6 ANALYSIS OF HYPERPARAMETERS

For KMD-Ensemble, we used 5 models in the ensemble, as we found this to provide good performance without significantly increasing space or computation requirements. Further details on the effect of different ensemble sizes (number of models) on performance can be found in Appendix A. KMD itself has two key hyperparameters: transform dimension $D$ and the kernel scale $\sigma$. Figure 1 shows how these hyperparameters affect accuracy across different datasets. $D$ controls the balance between the memory usage and the accuracy of kernel approximation. We found that setting $D$ to 5000 provides a good balance, offering sufficient accuracy without excessive memory usage. $\sigma$ affects the scale of the RBF kernel and thus influences the separation of the transformed features. We fixed $\sigma$ for all the datasets after determining it for each backbone (see Apendix A). KMD performs well across all datasets with this parameter setting. This indicates that KMD can learn various tasks incrementally without the need for major adjustments to its configuration.

## 6 CONCLUSION

A large body of literature exists on class incremental learning (CIL). Most existing methods focused on mitigating the CF by Fine-tuning an LM through direct parameter updates or by learning prompts or adapters, but these approaches are still prone to CF and limited attentions have been paid to ICS. The proposed method KMD deals with both problems and is fundamentally different from the traditional approaches. KMD only uses a fixed LM as a feature extractor. It leverages the *Radial Basis Function* (RBF) kernel to enhance the feature representation through *Random Fourier Features* approximation. This kernelized representation is then used to compute class means and a shared covariance matrix. The final classification is based on Mahalanobis distance. Our experiments show that KMD-Ensemble significantly outperforms existing baselines and, more importantly, achieves accuracy on par or better than joint Fine-tuning, which is regarded as the upper bound of CIL.

*Limitations:* The proposed method relies on the assumption that the LM contains sufficiently rich features for the CIL tasks in the target domain. If the LM's features are not well-suited to a specific domain, the accuracy of our method may suffer. A standard approach to address this is to fine-tune the general-purpose LM using a large domain-specific corpus before applying it to CIL.

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

# A   ADDITIONAL ABLATIONS AND IMPLEMENTATION DETAILS

| Dataset | E=1 | E=2 | E=3 | E=5 | E=10 |
|---|---|---|---|---|---|
| **CLINC** | $95.91_{\pm 0.68}$ | $96.04_{\pm 0.28}$ | $96.09_{\pm 0.20}$ | $96.62_{\pm 0.08}$ | $96.67_{\pm 0.20}$ |
| **Banking** | $92.23_{\pm 0.32}$ | $92.73_{\pm 0.15}$ | $92.83_{\pm 0.06}$ | $93.03_{\pm 0.06}$ | $93.13_{\pm 0.12}$ |
| **DBpedia** | $94.13_{\pm 0.32}$ | $94.40_{\pm 0.20}$ | $94.43_{\pm 0.25}$ | $94.53_{\pm 0.12}$ | $94.97_{\pm 0.06}$ |
| **HWU** | $87.27_{\pm 1.39}$ | $89.34_{\pm 0.44}$ | $89.53_{\pm 0.05}$ | $89.78_{\pm 0.09}$ | $90.09_{\pm 0.19}$ |

Table 4: Final accuracy (%) of KMD-Ensemble across different ensemble sizes. E represents the number of models used in the ensemble.

| Backbone | $\sigma$ |
|---|---|
| paraphrase-MiniLM | 1e-2 |
| BART-base | 1e-4 |
| BERT-base | 5e-3 |
| RoBERTa-large | 5e-3 |
| T5-3b | 5e-2 |
| Mistral-7B | 5e-6 |

Table 5: The $\sigma$ value used in our experiments for each LM. We fix the value across all datasets after determining the optimal value.

