# OpenReview forum: "Language Models as Feature Extractors for Accurate Continual Learning"
_ICLR.cc/2025/Conference — ICLR 2025 Conference Withdrawn Submission_

### Official Review · Reviewer_kqDF · 2024-10-31

**Soundness:** 3
**Presentation:** 3
**Contribution:** 2
**Rating:** 5
**Confidence:** 4

**Summary:**

This paper proposes an approach called Kernel Mahalanobis Distance (KMD) to address two significant challenges in Class Incremental Learning (CIL): Catastrophic Forgetting (CF) and Inter-task Class Separation (ICS). KMD leverages a frozen pre-trained language model as a fixed feature extractor and employs Mahalanobis Distance for classification. To further enhance performance, the authors introduce two additional techniques: the Radial Basis Function (RBF) and an ensemble method for the final classification. With these combined strategies, the KMD-ensemble outperforms existing baselines and achieves impressive performance that approaches that of joint training.

**Strengths:**

- Introducing Mahalanobis Distance (MD) and kernel functions seems to be an effective strategy for enhancing performance in Class Incremental Learning (CIL).
- To address the computational challenges posed by the RBF kernel, the use of Random Fourier Features (RFF) makes the kernelization process feasible for CIL, enabling efficient calculations of class means and updates to the covariance matrix.
- The validation of KMD across various language models demonstrates its versatility and potential for real-world applications, effectively overcoming a common barrier faced by existing CIL methods.

**Weaknesses:**

- The methods employed in KMD are all existing techniques, yet there is a lack of analysis regarding why these methods were chosen and how they contribute to the model's effectiveness. Additionally, there is no discussion on how Inter-task Class Separation (ICS) improves after adopting KMD, despite ICS being a primary challenge the approach aims to address.
- The model does not appear to be very efficient. In Section 5.5.4, the authors analyze the memory usage of KMD, which does not demonstrate any clear advantages. More analysis regarding temporal efficiency should also be included.
- It seems strange that simply using Nearest Class Mean (NCM) can achieve outstanding performance over state-of-the-art approaches. Could there be more discussion on this aspect?

**Questions:**

- Why does NCM achieve such a high performance?
- Can there be more discussion about the efficiency, not only the parameters?
- The method directly uses one shared covariance matrix for all classes, what if we use individual covariance matrices for each class?
- The results on CLINC surpassing those of joint training are indeed perplexing. It might be worthwhile to set up a new joint experiment specifically for the KMD method, using KMD's classification approach for joint training. This could help explore its performance regarding forgetting, rather than just comparing accuracy. Such an investigation would provide a more comprehensive understanding of KMD's advantages and how it differs from conventional joint training methods.

---

### Official Review · Reviewer_aSq9 · 2024-11-01

**Soundness:** 4
**Presentation:** 3
**Contribution:** 3
**Rating:** 6
**Confidence:** 4

**Summary:**

This paper addresses the text classification problem within the context of class incremental learning (CIL). The authors utilize a fixed language model without any prompts or adaptors and calculate class means and covariances for classification through Mahalanobis Distance. The authors then introduce Kernel Mahalanobis Distance (KMD), which enhances the discriminative ability of data features by a kernel function, and further incorporate an ensemble mechanism to improve performance. Experimental results demonstrate the effectiveness of the proposed method.

**Strengths:**

+ The method is simple but effective, which is easy to follow. The proposed method does not introduce too much calculation burden but performs well in text classification under CIL.
+ The power of NCM and MD has been demonstrated in image classification under CIL to some extent. However, the experimental results in this paper are impressive.
+ The paper is well-organized and the description is clear.

Overall, I have a positive view of this paper and will consider raising my score if the authors effectively address my concerns in the weakness section.

**Weaknesses:**

+ Lack of comparison and discussion between the proposed method and RanPAC [1]. The motivation of KMD is similar to RanPAC, which also aims to project data features into a higher dimensional space to increase discriminative ability. Therefore, it is essential to compare KMD with RanPAC fairly, such as using the same fixed backbone with or without ensemble.

+ The paper would benefit from a more comprehensive analysis of prior methods. While the authors assert that “the features of LMs are well-suited …, and the key lies in how to utilize these features appropriately,” it is essential to identify which components could potentially hinder the performance of LMs. Alternatively, a more in-depth theoretical discussion on how the proposed method mitigates catastrophic forgetting or improves inter-task class separation would enhance the overall argument.

+ I wonder if the proposed method can be directly used in computer vision tasks. If so, how is its performance?
---
**Reference:**

[1] McDonnell, Mark D., et al. RanPAC: Random Projections and Pre-trained models for Continual Learning. Advances in Neural Information Processing Systems 36 (2024).

**Questions:**

Please refer to the weakness section.

---

### Official Review · Reviewer_1Mbu · 2024-11-02

**Soundness:** 3
**Presentation:** 3
**Contribution:** 2
**Rating:** 5
**Confidence:** 3

**Summary:**

This paper proposes avoiding catastrophic forgetting (CF) through a fixed model and reduces ICS by measuring Mahalanobis distance with enhanced features using kernel functions. In the experiments, the authors claim that the proposed method achieves performance comparable to or even better than joint fine-tuning.

**Strengths:**

The paper is well-organized and easy to follow.
The experimental results demonstrate promising performance.
Combining the kernel function with a distance-based classifier is a reasonable approach.

**Weaknesses:**

1. My main concern is the rationale for freezing the network. This prevents the model from adapting to new data, turning incremental learning into merely an update of inference rules (prototypes) rather than the model’s knowledge. The effectiveness of this approach depends on the alignment between pretrained knowledge and downstream tasks, which may not hold in real-world scenarios. Significant gaps between incremental tasks and pretrained knowledge would still require model parameter adjustments.
2. This also raises concerns about the fairness of the experimental evaluation. Table 1 shows that simple NCM and MD approaches achieve performance close to the upper bound, suggesting strong generalization from the pretrained model. This reduces the necessity of incremental learning on these datasets. The authors should consider applying incremental learning to tasks that pose more generalization challenges.
3. Lastly, the methodological novelty is limited. Classification using MD is common, enhancing it with kernel functions is also a standard practice. Therefore, the innovation of this method is limited.

**Questions:**

see weakness

---

### Official Review · Reviewer_5xU8 · 2024-11-03

**Soundness:** 3
**Presentation:** 3
**Contribution:** 3
**Rating:** 5
**Confidence:** 4

**Summary:**

The paper proposes an approach for class incremental learning (CIL) that leverages language models as feature extractors. The proposed approach applies a kernel-based method (Ensemble of Kernel Mahalanobis Distance) on a frozen language model representation to assign class labels based on existing labeled data. Empirical results on  NLP classification tasks demonstrate the proposed approach is free of forgetting and even outperforms the joint-training baseline.

**Strengths:**

The paper is well-written and the proposed approach is well-motivated: the use of kernels with random fourier features and its advantage over simple nearest class mean.

Empirical results demonstrate competitive performance on classification tasks (CLINIC, Banking, DBpedia, HWU), outperforming all baselines and joint fine-tuning. The findings underscore the impact of language model feature extractor. Interestingly, the proposed approach has negative forgetting and positive transfer.

**Weaknesses:**

Overall, I'm impressed by the strong empirical performance of the proposed approach. However, I see the reliance on pre-trained language models and the realism of the CIL setup as the main weaknesses of this paper.

**1. Reliance on Pre-trained Models and Realism of the CIL Setup**

The paper depends on the availability of a pre-trained language model, with the assumption that this model provides sufficient discriminative power to separate classes in the CIL task. The findings suggest that the pre-trained model has the knowledge required for the tasks, reducing the continual learning challenge to already separable features. I think this setup lacks realism, as CIL should ideally focus on teaching new knowledge to the model, e.g., different types of tail knowledge as tasks. It is unclear when to use the proposed pre-trained model as feature extractor and when to apply fine-tuning to teach new knowledge. I wonder would it defeat the purpose of continual learning if the model already has that knowledge.

**2. Upper Bound Baseline**

The experiments refer to joint fine-tuning over the full language model as the upper bound baseline. I believe a stronger upper bound would be the proposed ensemble of kernel Mahalanobis distance, based on the full training data. Fine-tuning the parameters of language models introduces technical complexities that might impact performance, especially if the model already has the knowledge.

**Questions:**

Please address the weakness section.

---

### Note · Authors · 2024-11-18

I have read and agree with the venue's withdrawal policy on behalf of myself and my co-authors.